# Inferring Vertical Tree Growth Direction of *Samanea saman* and *Delonix regia* Trees with the Pattern of Lateral Root Distribution Using the Root Detector

**Mohamad Miftah Rahman [1], Yoga Fredisa [2], Dodi Nandika [1], Naresworo Nugroho [1], Iskandar Zulkanaen Siregar [3] and Lina Karlinasari [1,*]**

1 Department of Forest Products, Faculty of Forestry and Environment, IPB University, Jl. Lingkar Akademik, Darmaga, Bogor 16680, West Java, Indonesia

2 Natural Resource and Environmental Management Science, Graduate School, IPB University, Kampus IPB Baranangsiang, Bogor 16680, West Java, Indonesia

3 Department of Silviculture, Faculty of Forestry and Environment, IPB University, Jl. Lingkar Akademik, Darmaga, Bogor 16680, West Java, Indonesia

* Correspondence: karlinasari@apps.ipb.ac.id

**Abstract:** The root system is important for supporting tree growth and stability. In this study, we analyzed the relationship between the main lateral root distribution pattern and vertical tree growth direction based on root detection and analysis of tree morphometry. Tree growth represented by morphometric data were measured directly, and the root distribution was identified using a sonic Root Detector. Sixteen targeted trees (eight *Samanea saman* and eight *Delonix regia* trees) in an urban area landscape were selected in this study. The Root Detector revealed that the average sonic velocity of lateral roots was 676.88 m·s$^{-1}$ for *S. saman* and 865.32 m·s$^{-1}$ for *D. regia.* For root distribution, Root Detector determined the average numbers of main lateral roots for *S. saman* and *D. regia*, which were 6 and 10, respectively. Based on correlation analysis, significant relationships were found between tree root sonic velocity and the degree of lean, height, and diameter of the tree; meanwhile the relationship between crown diameter and slenderness were not significant. Findings confirmed that, in relation to the root distribution and the growth direction of the trunk and crown, the lateral root is mainly distributed in the opposite direction of the tree lean rather than crown growth direction.

**Keywords:** tree stability; tree morphometric; acoustic root detector; root sonic velocity; tree crown

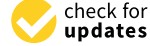



## 1. Introduction

Tree stability is related to the root system's vertical and lateral distribution. Tree roots grow as deep as the soil type, oxygen levels, and available moisture belowground. Tree root systems have three functions, including anchorage or structural stability, absorption of water and nutrients, and storage of vital food reserves [1,2]. Coarse roots (diameter > 2 cm) are especially important for structural stability and water, starch, and carbohydrate storage [3]. The theory of mechanical tree formation, as explained by Ylinen in 1952, mentioned that there were relationships between the physiological and mechanical systems of trees which were affected by several factors, such as the tree crown area and form, bending strength of wood, variation in bending strength, modulus of elasticity of standing timber, weight of the stem and crown, and size of the root system [4]. The failure of urban and forest trees is often a consequence of poor root condition caused by decay, injury, breakage, or low biomechanical acclimation to, for example, wind loads. Tree instability caused by root problems leads to threats to property and people in urban forests where a tree's root system, consisting of root distribution and directionality, root type, and root depth, is considered as an important factor.

However, evaluating the belowground root condition is not easy. There are many approaches for evaluating tree stability based on root systems. These approaches mainly

rely on visual assessment, tree morphometric measurements, and nondestructive testing that enables evaluating root system development, which are affected by site conditions [5,6]. The most common way to estimate the allocation to roots on a whole tree or stand level is through the development and application of allometric relationships. These relationships are defined according to parameters that are both easy to measure, such as tree diameter and height, and difficult to measure, such as stem, branch, foliar, below-stump, taproot, and lateral root biomass [7–9]. A complete belowground harvest of a single tree is difficult owing to multiple nearby neighbors and overlapping lateral root systems, although predefined pits proportional to tree size can be excavated at the base of the tree to capture the bulk of roots [10–12]. However, such methods are extremely time consuming and labor intensive, and they are not always an option, especially in urban environments, because of the presence of pavement and underground service lines, pipes, rocks, and other buried materials. Alternative methods for evaluating the root system in the field include using a pulling test, applying acoustic techniques to detect roots [13–15], and conducting detailed assessments with ground-penetrating radar to find the distribution of roots, especially coarse roots, below stumps [16,17]. This latter method can identify both taproots and lateral roots.

The acoustic root detection technique is based on an evaluation of the soundwave velocity differences in wood and soil. The technique can discriminate between pure soil and soil containing roots based on the path of the acoustic signal, and the results have been found to correlate with the safety factors predicted by pulling tests [5]. The limitations of the method are that it can only detect roots larger than 4 cm in diameter, and that the maximum measurement depth is about 0.5 m.

In the current study, we undertake root detection using an acoustic tool. Root mapping based on assessing root distribution with acoustic root detectors has previously been examined by Divos et al. [18], Proto et al. [1], and Rahman et al. [15]. These authors reported successfully identifying the lateral root distribution pattern. Several studies reported that acoustic root detection can be used to investigate the presence of radial root distribution with a maximum radial distance of approximately 6–8 times the diameter at breast height (dbh), with more than 80% of the root system being efficiently detected [19,20]. Additional information that has been noted is the optimal distance for root detection from the tool to the main trunk (i.e., 80–120 cm) and a significant correlation between root biomass and sonic speed within 30 cm of soil depth [1,5]. Our study aimed to analyze and infer the relationship between lateral root distribution pattern and vertical tree growth direction including tree morphology.

## 2. Materials and Methods

### 2.1. Sampling and Site Description

The study site was located in a business district in central Jakarta (latitude 6°13.5649′ S, longitude 106°48.523′ E). Two dominant urban tree species were selected for this study: rain trees [[*Samanea saman* (Jacq.) Merr.] and flamboyant trees [*Delonix regia* (Bojer ex Hook.) Raf.]. A total of 16 trees were selected, with eight trees from each species. The selection of individual trees was based on the tree diameter and the feasibility for root detection by a root detector. The diameter of each targeted tree was more than 30 cm, and the area within about 100 cm of the trunk was clear and open.

### 2.2. Tree Morphometry Analysis

The morphometric characteristics of the tree were analyzed based on the parameters used in a previous study conducted by Karlinasari et al. [6]. The dbh and tree height (h) were measured, and the data were then used to calculate the slenderness (S) coefficient as the ratio of tree height to dbh (S = h/d). The mean crown diameter (DCR) was the average of the longest and shortest crown diameters measured in eight subcardinal directions. The angle at which tree leaning was measured by Rangefinder Nikon Forestry Pro II.

### 2.3. Root Detection Measurement

The acoustic root detection tool used in the study was a Fakopp Root Detector (Fakopp Enterprises Bt, Hungary), which functions based on time-of-flight measurement. The tool consists of sensors, including a transmitter and a receiver, which are used to measure the propagation time of an acoustic signal through a material. The transmitter is a needle-like probe that is inserted into the trunk of a tree at ground level, and the receiver is a long metal spike (30 cm or longer) that has a good coupling for the soil. Both sensors are inserted at a position of about 45°.

The root detection is based on the difference in the speed of the acoustic signal between the roots' woody biomass and the soil through which signal propagates; differences in the density of the materials results in differences in the acoustic signal, which can then be used to determine the presence or absence of coarse roots. The transmitter sensor is hit by a hammer to generate a sound wave; the wave then propagates and is detected by the receiver sensor. Finally, the time between sound wave generation and detection is recorded. Based on the timing of wave propagation, the root detector tool can identify the presence of roots in the ground. The travel time of the signal decreases significantly when roots are detected within 10 cm of the receiver sensor. Measurement is carried out at 15 cm intervals along a circle around the trunk at a distance of 80 cm from the midpoint of the main trunk (Figure 1a). A distribution is then constructed based on the difference the acoustic signal value in both the soil and the root woody biomass.

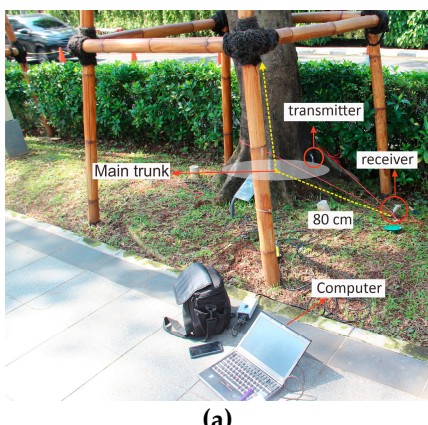
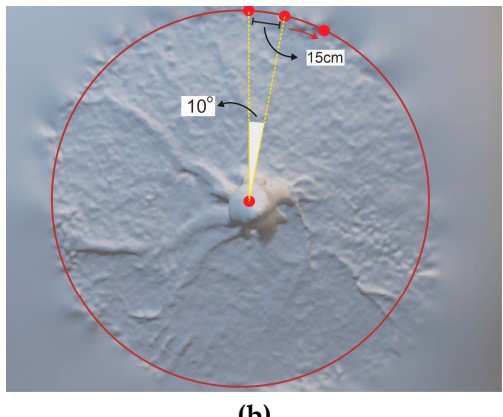

**(a)**      **(b)**

**Figure 1.** An experimental setup in the field for root detector testing: (**a**) Fakopp Root Detector set in the tree, and (**b**) 15 cm steps in a circle at a distance of 80 cm from the midpoint of the main trunk.

In this study, testing was carried out starting at a point directly north of the tree. For the distance to remain constant between the sensors, they were attached by a rope, and the sensor (piezo) was moved sequentially clockwise to trace a circle around the trunk in 15 cm steps from 0° to 360°. Each 15 cm step was at a distance of 80 cm from the midpoint of the main trunk, and the magnitude and direction of the sonic speed were measured for each 10° sector (Figure 1b) to obtain 34 observation points. The setting was in line with the study of Proto et al. [1]. Measurements were repeated three times at each point. The data were then recorded for analysis with the Root Detector evaluation application (Fakopp Enterprise Bt, Hungary) to produce sonic velocity data.

### 2.4. Root Architecture Analysis

The tool primarily detected the shallow and coarse roots [15]. The lateral distribution of roots was plotted based on the Root Detector Evaluation Software (Figure 2a). Based on reference values, the acoustic signal of soil was in the range of 200–400 m·s$^{-1}$ [5], while root woody biomass was detected for more than 400 m·s$^{-1}$ and occasionally ranging between 2000 and 4000 m·s$^{-1}$ [5]. The latter values varied depending on several environmental factors, especially the soil type and characteristics, as well as the humidity [1,5]. The Figure 2a image displays a circle symbol representing the intensity of black color based on

sonic velocity values. A darker circle indicates a higher velocity value. Based on previous research, 400 m·s$^{-1}$ has been used as a reliable threshold root presence [5]. Therefore, velocity above the threshold is considered a root. However, in a large tree which has various sizes of roots, root distribution is commonly determined by the main structural roots. It is possible that 2–3 observation points detected a main root due to those roots' size. Consequently, further analysis is required to differentiate a main root, as shown in Figure 2b. In determining the main root, it is necessary to find the mean of the sonic velocity values for each tree which were obtained from the velocity values above 400 m·s$^{-1}$. Main roots are detected as a peak value. The peak is flanked by two lower values, but remains relatively higher due to root diameter interference. The number of main roots is determined based on the number of peak sonic velocities greater than the mean velocity of all roots to eliminate bias from interference near test points and other environmental factors.

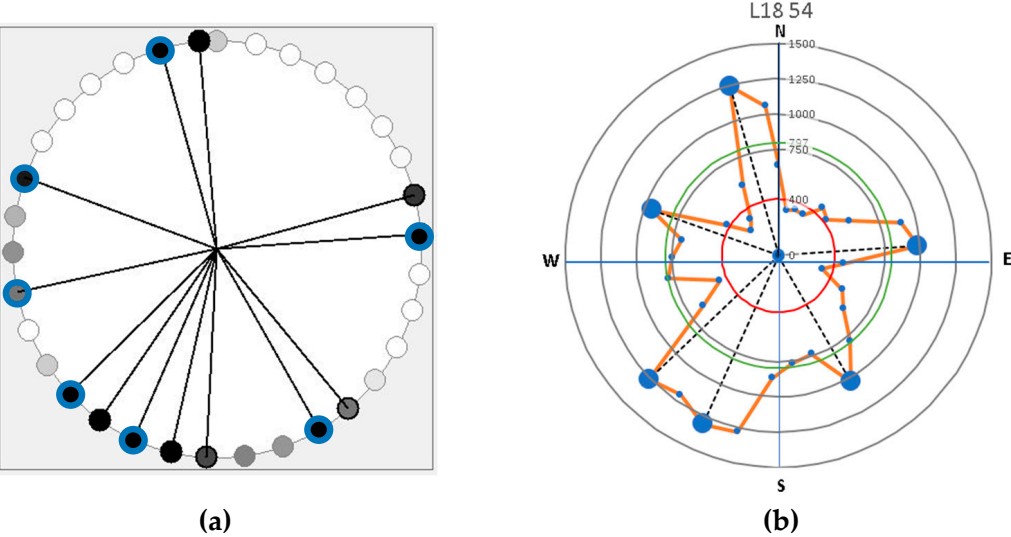

**Figure 2.** The example of representative graphical visualization of root distribution based on (**a**) Root Detector Evaluation Software (Fakopp Enterprise Bt, Hungary) in which dark circles points out a higher sonic velocity values of main root; and (**b**) sonic velocity and distribution data processing using Microsoft Excel (red line circle is root sonic velocity threshold, green line circle denotes the mean of root sonic velocity value, and the bigger peak dots indicate the main lateral root).

The lean of the tree, which followed the direction in which the crown tended to grow, was measured to obtain the relationship between the distribution of lateral roots belowground and the direction of tree growth. This information was overlaid onto the graph of the root distribution from the root detector analysis for each tree (Figure 3a). The tree stood upright at the midpoint, leaning in a clockwise direction, with 0° meaning a lean towards the north, 90° towards the east, and so on (Figure 3b). The magnitude of the tree lean and the crown direction tendency were measured and recorded in the morphometric analysis. In earlier studies, lateral roots have been shown to extend in the opposite direction of the crown load direction [21,22]. In the direction of the root zone, there is a minimum distribution area of about 30 degrees, particularly in the area facing the wind direction [22–24]. Although roots may not be detected in the direction of the tree crown load, that does not mean there is no root—especially if there is a vertical root (sinker root).

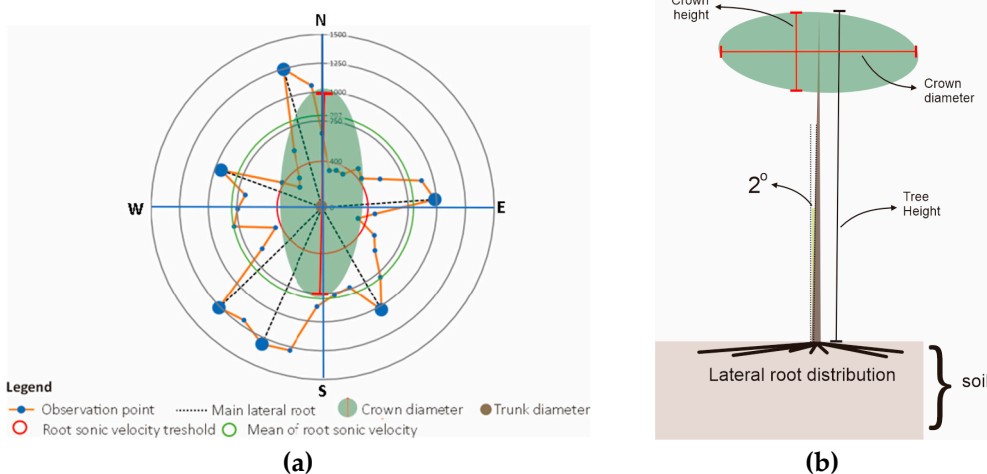

(a)                      (b)

**Figure 3.** Representative images of (**a**) the canopy area (green area) and the distribution of the main lateral roots based on the peak of sonic velocity (dash line), and (**b**) tree leaning and the crown direction tendency, which were related to the lateral root distribution and the direction of tree growth.

*2.5. Statistical Analysis*

The average values of each parameter for each tree were calculated and analyzed statistically. A software of IBM SPSS Statistics 25 (SPSS Inc., Evanston, IL, USA) was used to determine the relationship among morphometric parameters using correlation analysis (Spearman's test).

**3. Results and Discussion**

This study was conducted in an urban landscape where root system development may be limited due to structural barriers. The target tree species, rain trees (*S. saman*) (Figure 4a) and flamboyant trees (*D. regia*) (Figure 4b), both have a decurrent tree growth habit. The average tree dbh of *S. saman* and *D. regia* was 58.75 cm and 45.06 cm, respectively (Figure 5a). Meanwhile, the average tree height of *S. saman* was 15.25 m, and that of *D. regia* was 12.88 m (Figure 5b). Overall, the tree diameter and height for all study trees yielded an average dbh of 51.91 cm, while the average tree height (h) was 14.06 m. There was a significant relationship between dbh and the tree height of the two tree species (r = 0.702), as shown in Table 1.

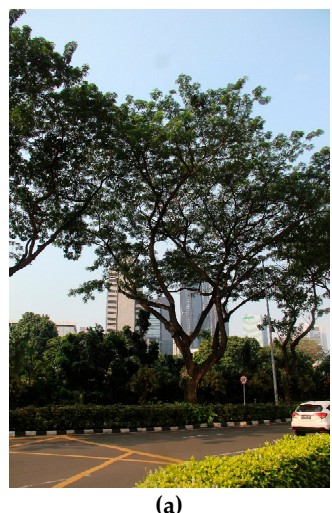 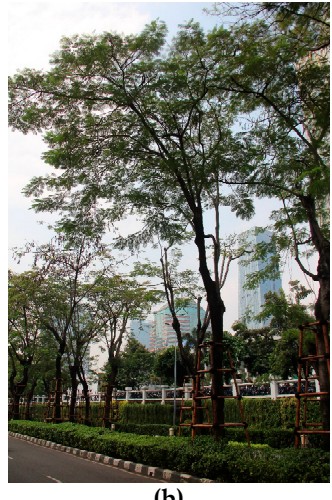

(a)                      (b)

**Figure 4.** The example of tree appearance of (**a**) rain tree (*Samanea saman*) and (**b**) flamboyant tree (*Delonix regia*) in the urban landscape.

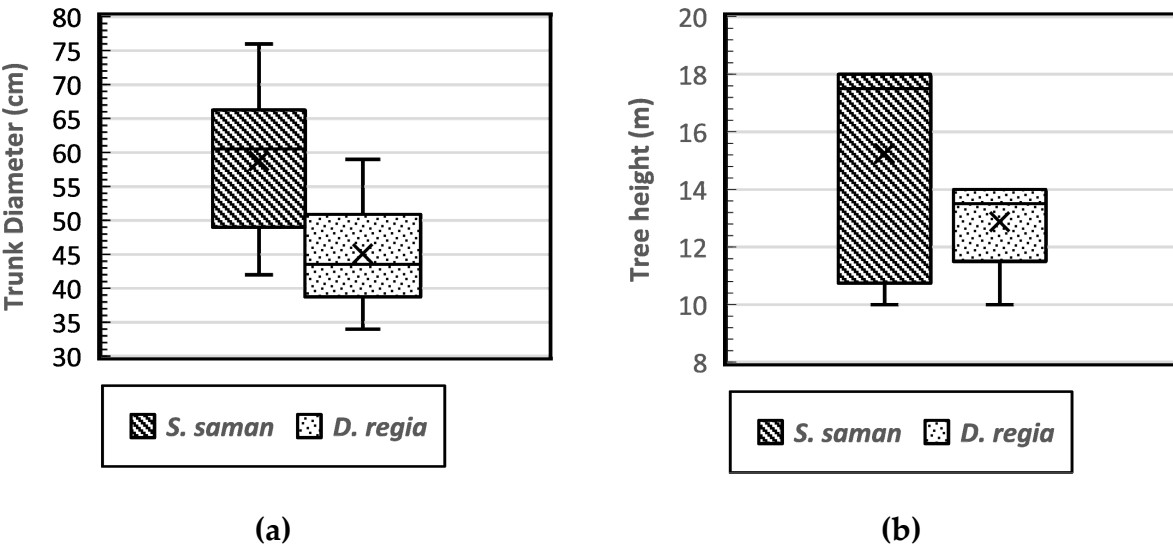

**(a)** **(b)**

**Figure 5.** Boxplot of the distribution of tree diameter (dbh) (**a**) and tree height (**b**) of *Samanea saman* and *Delonix regia*.

**Table 1.** Pearson's correlation for analysis of tree morphometric parameter and tree root sonic velocity (n = 16).

| Parameters | Trunk Diameter | Tree Height | Slenderness Coefficient | Crown Diameter | Tree Leaning Degree | Tree Root Sonic Velocity |
|---|---|---|---|---|---|---|
| Trunk diameter | 1 | | | | | |
| Tree height | 0.702 ** | 1 | | | | |
| Slenderness coefficient | −0.689 ** | −0.157 | 1 | | | |
| Crown diameter | 0.154 | 0.492 | 0.039 | 1 | | |
| Tree leaning degree | 0.661 ** | 0.545 * | −0.629 ** | 0.058 | 1 | |
| Tree root sonic velocity | −0.588 * | −0.726 ** | 0.389 | −0.016 | −0.747 ** | 1 |

** Correlation is significant at the 0.01 level (2-tailed). * Correlation is significant at the 0.05 level (2-tailed).

Several studies have reported a close relationship between dbh and height that is indicative of the stand stability, growth, and biomass characteristics [5,6,25,26]. The tree height ratio to dbh (h/dbh) is known as the slenderness coefficient, which serves as an indicator of tree stability. In a study of trees in urban areas, Mattheck et al. [27] found that a lower slenderness (<50) is associated with good tree stability, which is indicated by a larger crown, a lower center of gravity, and a better developed root system. Sharma et al. [28] also reported that lower slenderness values indicate that a tree has a greater crown length and a higher crown projection area, as well as a better developed root system, lower center of gravity, and higher stability. The results in the current study showed slenderness coefficients of *S. saman* and *D. regia* of 27.27 and 29.23, respectively (Figure 6a).

A spreading crown is shaped by lateral branches developing at the same level as the tree's main trunk with a decurrent or deliquescent form of growth habits, and with multiple dominant branches that do not exhibit significant apical dominance. Decurrent trees are based on a round to elliptic tree crown model [29]. The crown diameter of the two decurrent targeted tree species had an average length of 11.75 m and 10.41 m for *S. saman* and *D. regia*, respectively (Figure 6b). Rain trees (*S. saman*) have a characteristic dome-shaped canopy that is similar to an umbrella [30], while *D. regia* has a typical T shape that makes it well suited as a street tree [31]. Figure 6b shows that the distribution of the crown diameter of *S. saman* was more variable. Although the architecture of trees is determined by genetic factors that determine the development of the buds and branches of each species, trees in urban areas can be phenotypically influenced by the anthropogenic environment and may develop differently from those of the same species in an open area [32].

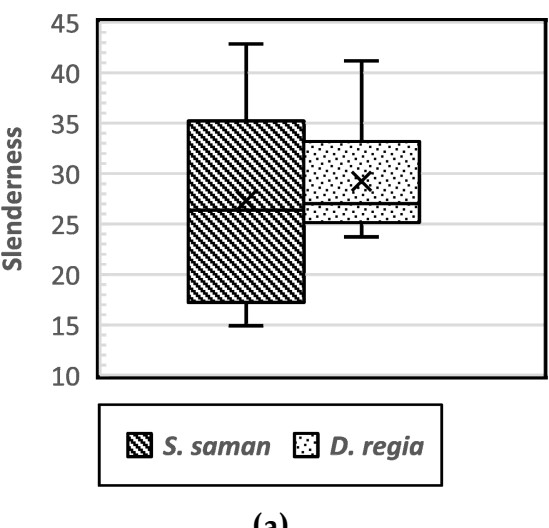

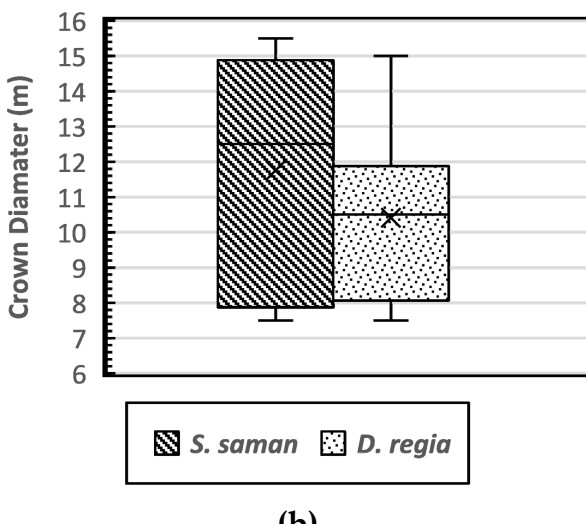

**(a)**

**(b)**

**Figure 6.** Boxplot of the tree slenderness (**a**) and distribution of crown diameter coefficients (**b**) of *Samanea saman* and *Delonix regia*.

Trees in urban landscapes are grown to beautify the streets and environment and to provide shade. Owing to the anthropogenic pressure in urban environments, the growth of trees along sidewalks and roads is limited by root access to water, air, and nutrients. As a consequence of these restrictions, trees may have rapid and disordered growth. Furthermore, root distribution varies among species under similar conditions. The behavior of root systems depends on species' environmental tolerances, including responses to soil characteristics.

In our study, sonic velocities were measured along circles around the trunk of trees to determine the lateral root distribution. The results showed that the sound wave velocity has a range value between 150 to 1800 m·s$^{-1}$ (Figure 7). The velocity of the acoustic signal in soil has previously been reported to be about 250–400 m·s$^{-1}$ depending on soil type and moisture content, while the sonic velocity in the roots can reach 2000–4000 m·s$^{-1}$ [18,33]. Based on those references, the Root Detector detected soil between 150 and 400 m·s$^{-1}$ and roots between 400 and 1800 m·s$^{-1}$. The values below 400 m·s$^{-1}$ were excluded from root analysis (Figure 8).

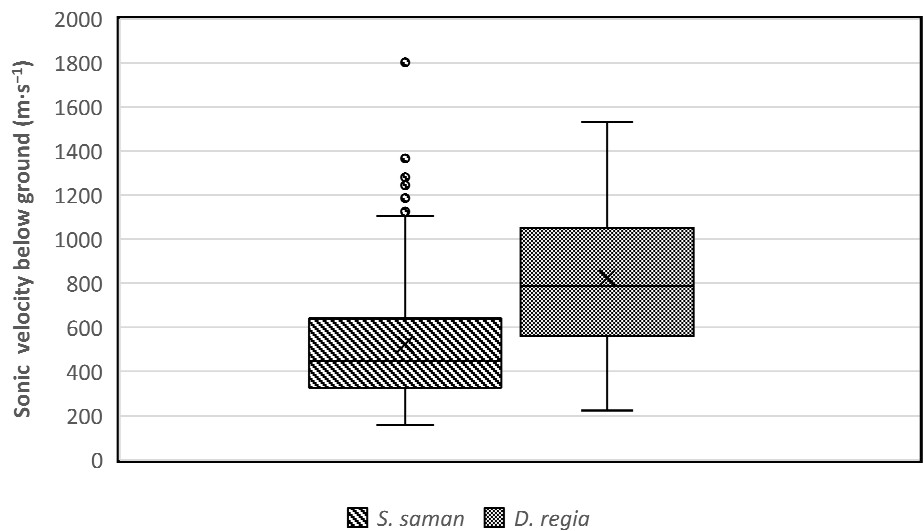

**Figure 7.** The sonic velocity (m·s$^{-1}$) detected by the Root Detector for *Samanea saman* and *Delonix regia*.

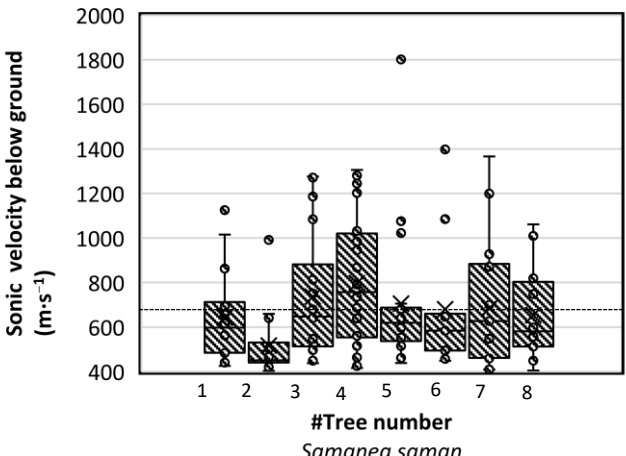 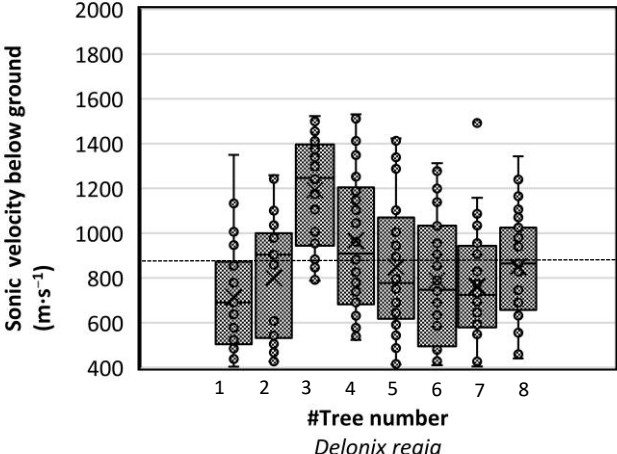

**Figure 8.** The root woody biomass sonic velocity (m·s$^{-1}$) detected by the Root Detector for *Samanea saman* and *Delonix regia* for all trees at about 80 cm distance from the trunk. The dash line (—) is the average value of root sonic velocity.

We found that the average soundwave velocities of the detected root woody biomass for the tree species *S. saman* and *D. regia* were 676.88 and 865.32 m·s$^{-1}$, respectively (Figure 8). These values were in line with a study by Proto et al. [1], who found the sonic velocity for olive trees to be around 500–1200 (m·s$^{-1}$). The sonic velocity of roots was previously reported to be affected by tree stem diameter, wood travel distance, and internal wood conditions, as well as tree age [34,35].

Tree growth and morphometric characteristics are often used to estimate root distribution [36]. However, in field conditions in urban settings, this is difficult to implement because of frequent cutting or inhibition of root distribution due to other physical impediments. Day et al. [2] reported that tree height and crown diameter are poor predictors of root distribution for several species in urban and landscape settings. Meanwhile, tree trunk diameter can be used to estimate the tree root spread under certain conditions, such as knowing the direction or growth, the angle at which the tree leans, and its location on a slope. The ratio of the root system radius to the trunk diameter was reported to be about 38:1 [2]. In summary, the root has a close relationship with the tree trunk diameter, which is strongly based on the species and environmental conditions.

The tree growth in the upper part of the trunk and the crown direction also affects the lateral root distribution belowground [2]. Our findings in Table 1 revealed a significant negative correlation between average root sonic velocity and tree trunk diameter, tree height, and tree leaning degree; however, a lower correlation was found between sonic velocity and the slenderness coefficient. The negative correlation means that bigger tree trunk diameter, higher tree height, and greater tree lean all generate lower root sonic velocity. This correlation is presumably due to the characteristics of the roots of bigger trees and/or trees with more lean will be deeper below the ground, so the signal interference due to the soil will be greater. Meanwhile, weak correlations were found between sonic velocity of tree roots and tree crown diameter. The crown diameter could be affected by anthropic activities such as pruning, while crown growth direction was related more to branch development instead of root system.

The importance of the root mechanical strength on anchorage is related to root geometry. The plant species, environmental conditions of soil moisture content and soil compaction level, and tree lean are presumably involved in root distribution, which is a means by which a tree adapts to its location. In a previous study, Stokes [21] explained that wind stress produced larger roots on the windward side of the tree relative to the other sides. Wind can increase the area of roots growing in the 30° sectors that face the wind direction. The presence of wind over time causes a tree to tilt, especially in the presence of the wind tunnel effect that often occurs in urban settings. Based on the roots'

acoustic signal, our study mapped root distributions related to the tendency of tree lean and tree crown direction (Figures 9 and 10, and Table 2). This study intended to clarify the symmetrical distribution of lateral roots in relation to the tree lean and canopy growth direction. Assessing the symmetry of the root system can reveal the preferred direction of biomass allocation. Root system symmetry was previously measured in terms of the center of mass of all the main lateral coarse roots [24]. Root mass can be represented by root diameter [37] or root cross-sectional area [19].

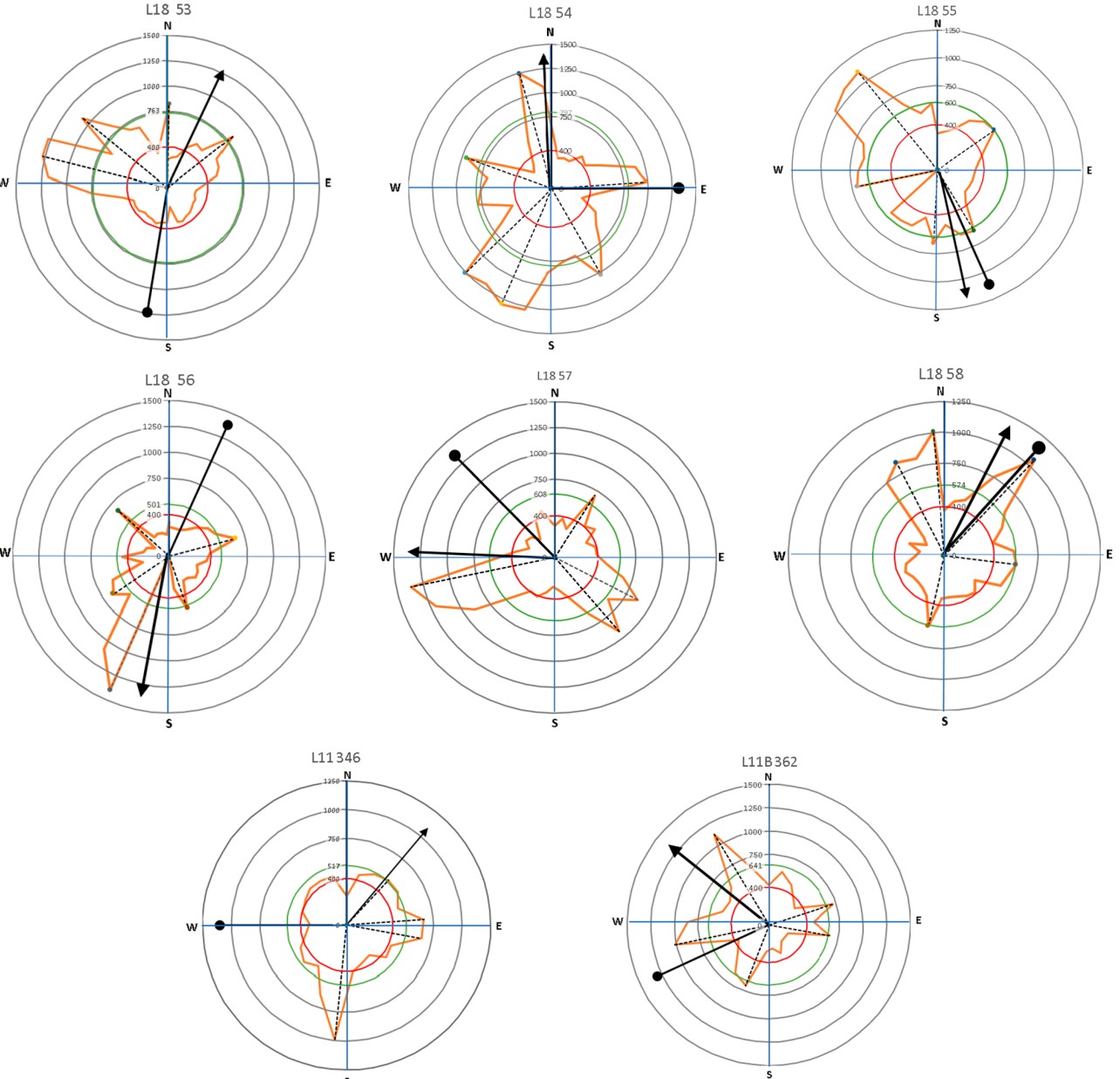

**Figure 9.** The direction of leaning for *Samanea saman* trees (circle-head line) and crown development (arrow-head line) in relation to lateral root distribution with the main lateral roots (dash lines).

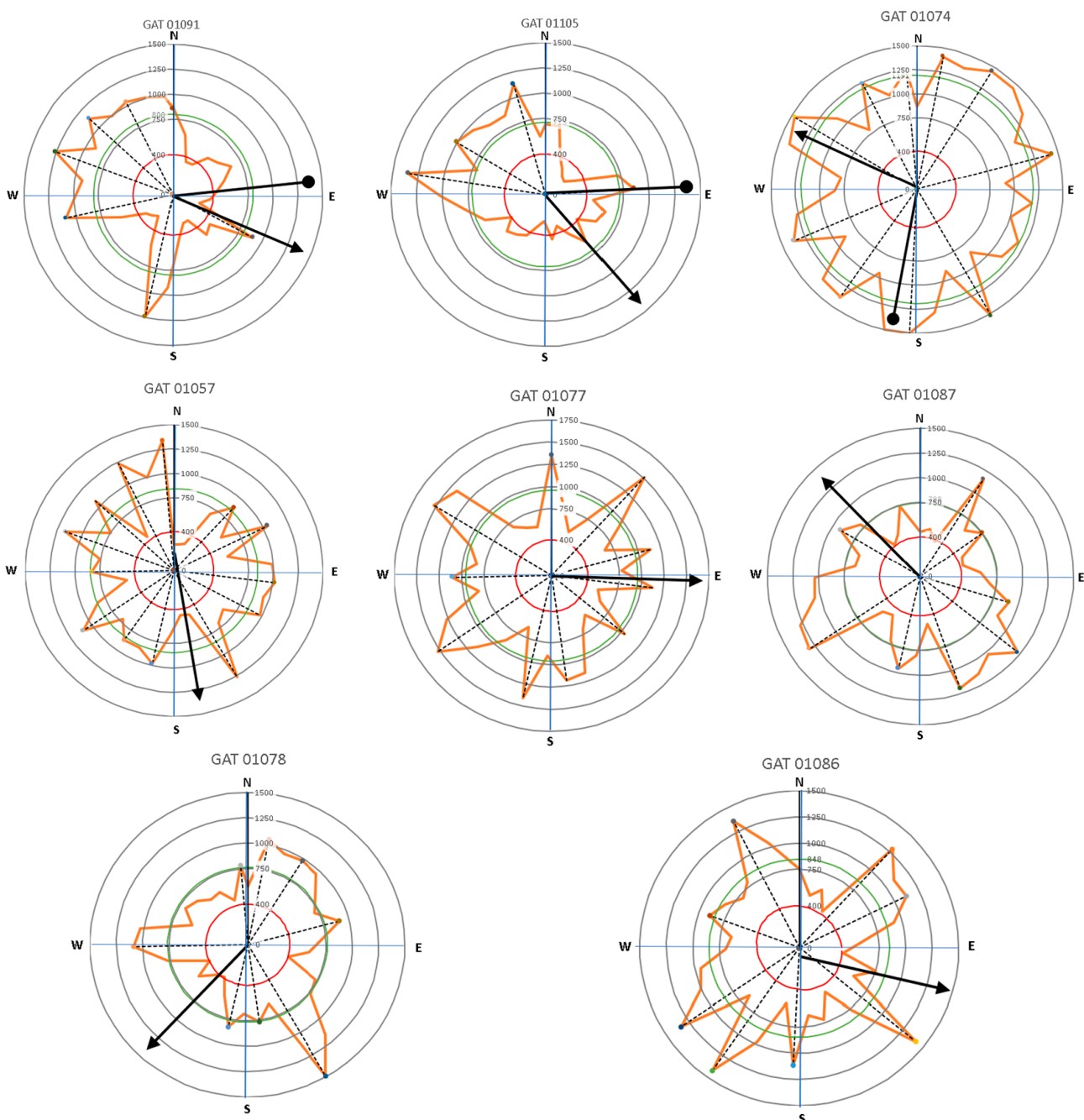

**Figure 10.** The direction of leaning for *Delonix regia* trees (circle-head line) and crown development (arrow-head line) related to lateral root distribution with the main lateral roots (dash lines).

**Table 2.** The direction of tree lean and crown development related to root lateral distribution.

| No | Species (Tree Code) | Tree Growth Direction | | |
| --- | --- | --- | --- | --- |
| | | Tree Lean (Magnitude) | Crown Direction | Root Distribution |
| 1 | *Samanea saman* (L18 53 80) | S (11°) | NE | NW |
| 2 | *Samanea saman* (L18 54 80) | E (2°) | N | SW |
| 3 | *Samanea saman* (L18 55) | SE (8°) | SE | NW & NE |
| 4 | *Samanea saman* (L18 56) | NE (10°) | SW | SW |
| 5 | *Samanea saman* (L18 57) | NW (20°) | W | SE & SW |
| 6 | *Samanea saman* (L18 58) | NE (15°) | NE | NE & NW |
| 7 | *Samanea saman* (L11 346) | W (25°) | NW | S |
| 8 | *Samanea saman* (L11B 362) | SW (17°) | NW | Distributed |
| 9 | *Delonix regia* (GAT 01091) | E (8°) | SE | NW |
| 10 | *Delonix regia* (GAT 01105) | E (10°) | SE | NW |
| 11 | *Delonix regia* (GAT 01074) | S (9°) | NW | Distributed |
| 12 | *Delonix regia* (GAT 01057) | Straight (0°) | S | Distributed |
| 13 | *Delonix regia* (GAT 01077) | Straight (0°) | E | Distributed |
| 14 | *Delonix regia* (GAT 01087) | Straight (0°) | NW | Distributed |
| 15 | *Delonix regia* (GAT 01078) | Straight (0°) | SW | Distributed |
| 16 | *Delonix regia* (GAT 01086) | Straight (0°) | SE | Distributed |

Notes: N (North), S (South), E (East), W (West).

Based on our results and the theoretical and reference values of root sonic velocity, the average sonic velocity values of *S. saman* were about 670 m·s$^{-1}$, and about 860 m·s$^{-1}$ for *D. regia*. The number of main lateral roots can be determined based on the velocity value peaks which exceed the average root sonic velocity values for each tree and the analysis of root distribution mapping. The identified root number in *S. saman* tree species ranged from 4 to 6, while *D. regia* had between 4 and 13 structural roots per tree (Figure 11). A study by Ramos-Rivera et al. [38] indicated that *S. saman* needs at least four large roots to provide stability for the tree. The higher number of roots for *D. regia* can be explained by a report from Woodward and Menninger [39], who found that *D. regia* has an extensive superficial root system, which renders it vulnerable to windthrow during storms. Tree species with shallow root systems are susceptible to attack by root rot and are liable to be uprooted during strong storms and broken by strong winds [38].

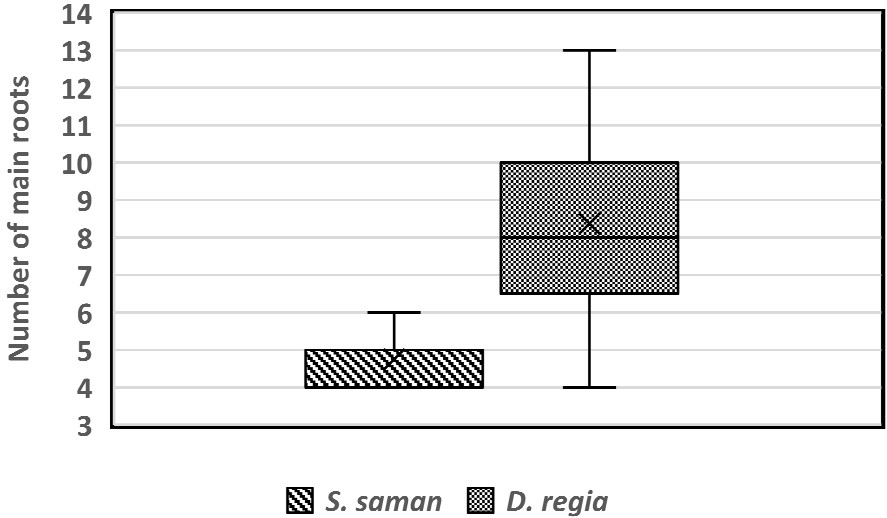

**Figure 11.** The number of main lateral roots detected for *Samanea saman* and *Delonix regia* trees.

Results presented in Figures 9 and 10 show that large roots indicated by high velocity are dominantly located opposite the tree lean direction, which are factors in the tree load rather than the crown (canopy) direction. Stokes et al. [21,22] and Chiatante et al. [40] mentioned that the distribution of roots was influenced by the environment, especially the

wind direction and soil slope. In addition, the lean of the trees tended to have a distribution that was centered in one direction. Meanwhile, symmetrical trees were more likely to have evenly distributed roots. An asymmetrical root distribution caused by mechanical force in the environment directly led to plant responses that increased the tree stability through root distribution [3].

## 4. Conclusions

The lateral root distribution of *S. saman* and *D. regia* were determined by the Root Detector and an analysis of tree morphometry. The root detector results showed that the lateral roots were detected at the average root sonic velocity of 670 m·s$^{-1}$ for *S. saman* and 860 m·s$^{-1}$ for *D. regia*. The root sonic velocity was significantly correlated with tree diameter, height, and leaning degree, but no significant difference with the crown diameter and slenderness coefficient of tree. *S. saman* had about 6 main lateral roots detected, and *D. regia* had about 10. A relationship existed between the lateral root distribution and the tendency of the tree growth direction. Overlays of the root distribution analysis with the tree lean and crown directions confirmed that the lateral roots were mainly distributed in opposite to the tree lean instead of crown direction. Our methods were based on the use of an acoustic root detector tool and can potentially be applied as a fast and simple way to detect the dominant direction of the lateral root distribution belowground in relation to vertical tree growth direction.

**Author Contributions:** Conceptualization, drafting, writing, L.K. review, editing, supervision, L.K., I.Z.S.; Field and laboratory work, data curation, M.M.R., Y.F.; Original draft preparation, L.K., M.M.R.; Supervision, I.Z.S., N.N., D.N.; Formal analysis, M.M.R., Y.F., L.K. All authors have read and agreed to the published version of the manuscript.

**Funding:** This research was funded by the Directorate General of Higher Education, Research and Technology, Ministry of Education, Culture, Research and Technology, Republic of Indonesia: 3628/IT3/PT.01.03/P/B/2022 and 001/E5/PG.02.00PT/2022.

**Data Availability Statement:** Not applicable.

**Acknowledgments:** The authors are grateful for the support of the Directorate General of Higher Education, Ministry of Education, Culture, Research and Technology, Republic of Indonesia, through Research Grants in scheme Fundamental Research, FY 2022 and PT Danayasa Arthatama. We also would like to thank Utami Syafitri for the valuable insights and discussion regarding the statistical analysis.

**Conflicts of Interest:** The authors declare no conflict of interest.

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
