# Peer review of "Inferring Vertical Tree Growth Direction of Samanea saman and Delonix regia Trees with the Pattern of Lateral Root Distribution Using the Root Detector"

_forests, doi:10.3390/f14020427_

Round 1

Reviewer 1 Report

The study focuses on the spatial distribution of coarse roots in the soil in urban areas. Here are some questions and comments that need clarification.

1. How can the slenderness coefficient indicate a better development of root systems?

2. The slenderness coefficient in your study was 27.27 and 29.23, or 027 and 029, as illustrated in Figure 6?

3. How deep can the device determine the roots based on the characteristics? And the minimum diameter? How deep have you been able to identify the roots in your research? And what is the minimum diameter of tree roots found?

4. Is the speed of the sound wave dependent upon the age of the trees? From the condition of the trunks (rotting, mechanical damage, etc.)? Why were those studies not conducted?

5. Was the root detection tool used as independent equipment or as an extension for the ArborSonic 3D package?

6. Table 2 is presented in graphical form, which is impractical for collecting information. Can you change the table into a figure?

Author Response

Thank you for your comments and suggestions.

  1. How can the slenderness coefficient indicate a better development of root system

Yes, it refers to Sharma et al. 2019 who mentioned: “Smaller values of Height to diameter at breast height ratio (HDR or slenderness) indicate longer crown length, higher crown projection area, better-developed root system, lower position of the center of gravity, and higher stability of the trees” (Sharma et al. 2019). It means that a good root system will optimize the absorption of nutrients from the soil, which allows the tree to get optimum nutrition for stem growth horizontally, resulting in a lower slenderness ratio.

We have added the statement in the text (part of the result and discussion, second paragraph) and put the reference in the list.

  1. The slenderness coefficient in your study was 27.27 and 29.93, or 027 and 029, as illustrated in figure 6

We have corrected the value in figure 6. It seemed the format of the graph (y-axis) was automatically changed, and it should be 27.27 and 29.93.

  1. How deep can the device determine the roots based on the characteristics? And the minimum diameter? How deep have you been able to identify the roots in your research? And what is the minimum diameter of tree roots found?

We refer to the study of Proto et al. 2020, as mentioned in the introduction (the third paragraph). The root detector can detect the root size at a 4 cm minimum, and the maximum depth of the measure is about 0.5 m based on excavation. In this study, we did not excavate the root system, so we do not have any information on the root characteristics (diameter and depth) of the targeted trees in this study.

  1. Is the speed of sound dependent upon the age of the trees? From the condition of the trunks (rotting, mechanical damage, etc.)? Why were those studies not conducted?

Yes, referring to Russo et al. (2020), the tree age has an effect on the speed of sound, and the higher tree age has a higher speed of sound. However, in this study, the tree age did not differ between the tree sampled based on the information from the tree manager. The condition of rotting and mechanical damage would also affect the speed of sound due to the difference in the density of the wood.

To explain the factors which affect the speed of sound on woody-biomass, we have added the statement and put the references in the text (Russo et al. 2020; Zhang et al. 2011)

Our study focused on the root lateral root distribution detected by the Root Detector, and we did not do root excavation to clarify the result.

  1. Was the root detection tool used as independent equipment or an extension for the ArborSonic 3D package?

The root detection tool used is independent equipment from ArborSonic 3D Package. The hardware and software are FakoppÒ Root Detector (https://fakopp.com/en/product/rootdetector/)

  1. Table 2 is presented in graphical form, which is impractical for collecting information. Can you change the table into a figure?

We have converted the table into figures, becoming figures 10 and figure 11.

Reviewer 2 Report

This is a well written paper. It overall reads well and I only have a few comments:

- Figure captions need to report the genus and species names in full.

- "Velocity of roots" sounds odd. Any other synonyms the Authors can use?

- This study only focused on the "rain tree" and the "flamboyant tree". Perhaps include this information in the title?

Author Response

Thank you for your comments and suggestions

  1. Figure captions need to report the genus and species names in full

We already put the genus and species names in each caption

  1. “Velocity of roots” sounds odd. Any other synonyms the authors can use?

“Velocity of root” already change to “tree root sound velocity.” The term is also stated in table 1.

  1. This study only focused on the “rain tree” and “flamboyant tree.” Perhaps include this information in the title?

We have accommodated the suggestion. The title would be: “Mapping of Tree Growth Direction of Samanea saman and Delonix regia trees Based on the Lateral Root Distribution Evaluated by the Root Detector.

Reviewer 3 Report

This is my first revision, and I enjoy it. This manuscript is well-written, easy to understand, and all important information were highlighted. Firstly, this is manuscript is in line with the MDPI Forests journal. In this study, authors analyzed root architecture in a high-urbanized urban area in Jakarta. In my view, this is an important topic for urban foresters, but not enough investigated, and should be processed. English is fluent and easy to read, and the data were well presented and easy to understand. Please note that I’m a plant scientist and I’m not relevant to elevating a technical part (reliability of used sensors etc.) of this study.

In my ‘biological’ view, this manuscript didn’t have difficulties and can be accepted in its present form.

Author Response

Thank you for your comments; we do our best for the manuscript.

Round 2

Reviewer 1 Report

I am pleased with the authors’ responses to the questions and remarks. The authors have revised the manuscript. On this basis, I believe this article can be accepted for publication.

Author Response

As we have answered before, comments have been submitted in response on November 20, 2022, through the system and reviewer 1 has provided a reply comment. Herewith, we convey again the response to the comments submitted in Round 1.

Thank you for your comments and suggestions.

  1. How can the slenderness coefficient indicate a better development of root system

Yes, it refers to Sharma et al. 2019 who mentioned: “Smaller values of Height to diameter at breast height ratio (HDR or slenderness) indicate longer crown length, higher crown projection area, better-developed root system, lower position of the center of gravity, and higher stability of the trees” (Sharma et al. 2019). It means that a good root system will optimize the absorption of nutrients from the soil, which allows the tree to get optimum nutrition for stem growth horizontally, resulting in a lower slenderness ratio.

We have added the statement in the text (part of the result and discussion, second paragraph) and put the reference in the list.

  1. The slenderness coefficient in your study was 27.27 and 29.93, or 027 and 029, as illustrated in figure 6

We have corrected the value in figure 6. It seemed the format of the graph (y-axis) was automatically changed, and it should be 27.27 and 29.93.

  1. How deep can the device determine the roots based on the characteristics? And the minimum diameter? How deep have you been able to identify the roots in your research? And what is the minimum diameter of tree roots found?

We refer to the study of Proto et al. 2020, as mentioned in the introduction (the third paragraph). The root detector can detect the root size at a 4 cm minimum, and the maximum depth of the measure is about 0.5 m based on excavation. In this study, we did not excavate the root system, so we do not have any information on the root characteristics (diameter and depth) of the targeted trees in this study.

  1. Is the speed of sound dependent upon the age of the trees? From the condition of the trunks (rotting, mechanical damage, etc.)? Why were those studies not conducted?

Yes, referring to Russo et al. (2020), the tree age has an effect on the speed of sound, and the higher tree age has a higher speed of sound. However, in this study, the tree age did not differ between the tree sampled based on the information from the tree manager. The condition of rotting and mechanical damage would also affect the speed of sound due to the difference in the density of the wood.

To explain the factors which affect the speed of sound on woody-biomass, we have added the statement and put the references in the text (Russo et al. 2020; Zhang et al. 2011)

Our study focused on the root lateral root distribution detected by the Root Detector, and we did not do root excavation to clarify the result.

  1. Was the root detection tool used as independent equipment or an extension for the ArborSonic 3D package?

The root detection tool used is independent equipment from ArborSonic 3D Package. The hardware and software are FakoppÒ Root Detector (https://fakopp.com/en/product/rootdetector/)

  1. Table 2 is presented in graphical form, which is impractical for collecting information. Can you change the table into a figure?

We have converted the table into figures, becoming figures 10 and figure 11.
